# Urban Open Space Is Associated with Better Renal Function of Adult Residents in New Taipei City

**DOI:** 10.3390/ijerph16132436

**Published:** 2019-07-09

**Authors:** Jien-Wen Chien, Ya-Ru Yang, Szu-Ying Chen, Yu-Jun Chang, Chang-Chuan Chan

**Affiliations:** 1Institute of Occupational Medicine and Industrial Hygiene, College of Public Health, National Taiwan University, Taipei 10020, Taiwan; 2Department of Pediatric Nephrology, Changhua Christian Children’s Hospital, Changhua 50006, Taiwan; 3Division of surgical intensive care, Department of critical care, E-Da Hospital, Kaohsiung 82445, Taiwan; 4Department of Nursing, Fooyin University, Kaohsiung 83102, Taiwan; 5Epidemiology and Biostatistics Center, Changhua Christian Hospital, Changhua 50006, Taiwan

**Keywords:** open spaces 2, CKD 3, renal function

## Abstract

The purpose of this study is to explore the association between proximity to open space and adult renal function. This was a cross-sectional study. Adult residents of Taipei metropolis were recruited in the analysis. The proximity of each subject to open space was measured using the Geographic Information System. Residents were divided into two groups: with and without chronic kidney disease (CKD). We made univariable comparisons between the two groups. The logistic regression models were used to estimate the odds ratio of CKD. Forest plot was used to examine the effect of interaction between distance to open space and subgroup variable on CKD. A total number of 21,656 subjects with mean age 53.6 years were enrolled in the study. Of the subjects, 2226 (10.28%) had CKD. The mean and standard deviation of distance to open space were 117.23 m and 80.19 m, respectively. Every 100 m distance to open space was associated with an odds ratio of 1.071 for CKD. Subgroup analysis revealed that residents of female, without hypertension, or without impaired fasting glucose (IFG) living more than 200 m from open spaces have greater odds of CKD than those living less than 200 m. Conclusions: Proximity to open space was associated with a lower prevalence of CKD among adults in Taiwan. Such association was enhanced among females and healthy adults without hypertension or impaired fasting glucose (IFG).

## 1. Introduction

It is well known that the ecosystem affects the well-being of humankind. However, urbanization is a global trend, especially in Asia. More than half of the world’s population currently live in cities [1]. Urban citizens living in artificial environments may lack the services providing by the ecosystem. Fortunately, open spaces can compensate for it. The effects of open space on health include both physiological and psychological benefits. Open space independently augments the physical activity of nearby residents [2]. The urban environmental design may also influence behavior, mental health [3] and well-being [4] of residents. Other health benefits include an increase in birth weight (grams) among the lowest education level group [5], decrease in number of small for gestational age births [6], lower prevalence of early childhood asthma [7], lower morbidity (especially for anxiety disorder and depression) [8], lower mortality (especially for respiratory disease) [9], lower heat-related mortality [10], increase in longevity of urban senior citizens [11], and lower risk of stroke mortality [12].

Studies on the health effects of open space in Asia are relatively fewer. Moreover, to our best knowledge, the association between open space and renal diseases has not been explored. In Taiwan, the prevalence of chronic kidney disease (CKD) among adults ranged from 6.9% [13] to 11.9% [14], exerting a huge burden on the health care system. This study aims to explore the relationship between proximity to open space and renal diseases among adults.

## 2. Materials and Methods 

Study Population: Residents aged 30 and above who lived in metropolitan areas (six districts with population densities higher than 20,000/km^2^) in 2009 were the target of the study (Figure 1). One of the districts, Yonghe City, has the highest population density in Taiwan, with population density 41,446/km^2^ in 2009. Subjects were recruited from the New Taipei City Health Screening Program from 2007 to 2009. This annual program was supervised by the Department of Health of the New Taipei City Government. All citizens aged 30 and above were invited to participate in the program every three years. Subjects with incomplete information were excluded from the study. Finally, 2630 subjects were excluded due to missing data and a total of 21,656 subjects from six districts were included in this study. Permission to use these decoded data from the Department of Health of the New Taipei City Government was granted. This study was approved by the Joint Institutional Review Boards of National Health Research Institutes (Approval number: EO-104-PP-09).

Health Data: The annual health screening program was implemented by the Department of Health of the New Taipei City Government. The program involved a self-reporting questionnaire, interview by a physician, and blood sampling for biochemistry study. Demographic data including age, gender, location of residence, alcohol consumption, smoking, and betel-nut-chewing habit were collected. During the health examination, weight, height, and blood pressure of each subject were measured. Serum creatinine was analyzed using isotope dilution mass spectrometry (IDMS)-traceable method. Hypertension was defined as a systolic blood pressure ≥140 mmHg or a diastolic blood pressure ≥90 mmHg. Overweight was defined as body mass index (BMI) ≥24 (kg/m^2^). Impaired fasting glucose (IFG) was defined as fasting sugar ≥126 mg/dL. Hypercholesteremia was defined as serum cholesterol level ≥200 mg/dL. 

Urinanalysis: Each subject’s first morning mid-void urine was collected during the day of the survey. Urinanalysis was performed using Multistix test strips (Bayer Diarnostics, Victoria, Australia) which grade proteinuria as negative, trace, 1+ (0.3 g/L), 2+ (1 g/L), 3+ (3 g/L) or 4+ (≥20 g/L) by a reaction with tetrabromphenol blue.

Renal Function: To estimate the glomerular filtration rate (GFR), the CKD-EPI-Taiwan equation was used. This equation has been proved to have a lower bias than that derived from the modification of diet in renal disease (MDRD) study [15]. The CKD-EPI-Taiwan equation is as follows: eGFR = 1.262 x [1.41xmin(Scr/κ, 1)^α^ x max(Scr/κ,1)^1.209^ x 0.993^Age^ x 1.018(if female) x 1.159(if black)] ^0.914^, where Scr denotes serum creatinine, κ is 0.7 for female and 0.9 for male, α is -0.329 for female and -0.411 for male, min indicates the minimum of Scr/κ or 1, and max indicates the maximum of Scr/κ or 1. In this study, CKD was defined as eGFR ≤60 mL/min/1.73 m^2^, which stands for 50% loss in renal function.

Distance to Open Space: Locations of subjects’ residence were geocoded using the ArcGIS 10.1 software, and their minimum distance to open space was calculated. Open space is defined by the US EPA as any open piece of land that is undeveloped (has no buildings or built structures) and is accessible to the public [16]. Examples of open space are green space, schoolyards, playgrounds, public plazas and vacant lots. The land use database was established by the National Land Surveying and Mapping Center, Taiwan in 2007. According to the standard land use coding manual of the database, open spaces include public utilities (schools, code 00202-0260204) and recreation areas (park, code 070201; playground, code 070202; and sports facility, code 070203) (Figure 2).

Particulate Matter Exposures: We used land use regression (LUR) model to estimate annual average concentrations of PM_2.5_, PM_2.5Absorbance_, PM_10_, and PM_Coarse_ at each resident’s address. The LUR model was developed for the European Study of Cohorts for Air Pollution Effects (ESCAPE) project [17]. To develop the model, we measured the pollutants in 20 different sites during three 14-day periods from October 2009 to August 2010. The detailed protocol of the model was described in the previous paper [18]. 

Statistical analysis: Patients were divided into two groups according to the level of eGFR: with and without CKD. The demographic data and other clinically relevant data of continuous variables are presented as mean and standard deviation, whereas categorical variables were presented as numbers and percentages. We made univariable statistical comparisons between groups by Student’s t-tests for continuous data, and chi-square tests for categorical data. The logistic regression model was used to estimate the odds ratio of CKD. Variables that achieved statistical significance in univariate analysis were subsequently entered into multivariate analysis. According to the distance from the open space, it is divided into two groups of ≥200 meters and <200 meters. Forest plot showing the effect of interaction between distance to open space and subgroup variable (age, gender, overweight, hypertension, impaired fasting glucose (IFG), age, smoking, and drinking) on CKD. The final model retains only statistically significant factors after multiple regression. All data were analyzed using the IBM SPSS Statistics for Windows, Version 22.0 (IBM Corp., Armonk, NY). *p*-values < 0.05 were considered statistically significant.

## 3. Results

New Taipei City had the most population among Taiwan’s administrative regions. The metropolitan area of the city includes six districts, which had population densities range from 20,308/km^2^ to 41,446/km^2^. The six districts had a total area of 924,855 km^2^ and total population of 2,190,426 citizens at the end of 2009. The distribution location of open spaces is shown in Figure 2. The recreational spaces (include parks, playgrounds and sport venues) had a total area of 6.46 km^2^ (6.99%). The school spaces (include elementary schools, high schools, and colleges) had a total area of 3.9km^2^ (4.22%). Per capita open space availability is 4.7m^2^.

Table 1 lists the demographic data of our study subjects. In brief, there were 21,656 subjects in total, with a mean age 53.6 years and male to female ratio of 1:2. About half (50.4%) of the subjects were overweight, and 33.2% of subjects were hypertensive. Moreover, 18.3% of subjects had smoked, 36.7% had consumed alcohol, and 3.5% had chewed betel nut. The mean eGFR was 77.05 mL/min/m^2^. The prevalence of CKD (eGFR < 60 mL/min/m^2^) was 10.3%. The mean ± SD distance to open space was 117.23 ± 80.19 m. The longest distance to open space was 570.7 m.

Study subjects were divided into two groups according to the level of eGFR: with and without CKD. We made univariable statistical comparisons between groups by Student’s t-tests for continuous data, and chi-square tests for categorical data (Table 2). We found that variables significantly associated with CKD include, age ≥65 years old, male gender, education level, smoking, alcohol consumption, hypertension, overweight, impaired fasting glucose (IFG), and proteinuria.

The logistic regression model was used to estimate the odds ratio (OR) of CKD. Variables with significant OR of CKD after univariate analysis were subsequently analyzed by multivariate method (Table 3). We found that distance to open space (every 100 m) has significant odds (1.071) for CKD. Other parameters also have significant odds for CKD include: PM_coarse_ (OR 1.017), age ≥65 (OR 6.812), male (OR 2.741), uneducated (OR 1.714), elementary or junior high school (OR 1.339), overweight (OR 1.308), hypertension (OR 1.184), cholesterol level (OR 1.002), lower red blood count(OR 1.876), higher leukocyte count (OR 1.097), proteinuria +/- (OR 2.156), proteinuria + (OR 5.645), proteinuria ++ (OR 3.701), and proteinuria +++ (OR 15.028). 

Forest plot (Table 4) showing subgroup analysis of the risk of CKD in different distance to the open space. The result shows that if residents of the female gender, without hypertension, or without impaired fasting glucose (IFG) living more than 200 m from open spaces, they would have greater odds of CKD than those living less than 200 m from open spaces.

Table 5 reveals the results of multiple linear regression on eGFR. Variables with significant odds of CKD were selected for this analysis. We found that predictor with the largest negative partial regression coefficient (β) was proteinuria (−4.770). The β value of distance to open space (100 m) and PM_Coarse_ were −0.185 and −0.122, respectively.

## 4. Discussion

The evidence of the beneficial effects of urban open space on human health is substantial. Prior research revealed that urban open space has many benefits on health. Maas et al. investigated the relationship between physician-assessed morbidity and green space in the living environment of residents [8]. They found lower morbidities caused by anxiety (OR: 0.95), depression (OR: 0.96), coronary heart disease (OR: 0.97), infectious disease of the intestinal canal (OR: 0.97), urinary tract infection (OR: 0.97), musculoskeletal disease (OR: 0.98), and diabetes mellitus (OR: 0.98) among people in residences having green space within 1-km radius. Other benefits of open space on the health of residents in the vicinity include lower blood pressure [19], less obesity [20], decreased mortality of respiratory disease [9], less heat-associated diseases, especially among elderly people with chronic diseases (congestive heart failure, myocardial infarction, chronic obstructive pulmonary disease, and diabetes mellitus) [10] , lower stroke mortality [12], increased longevity of senior citizens [11], less newborns with low birth weight [5,6], and less children with asthma [7].

The mechanisms behind beneficial effects of open space on physical and mental health include enhancing physical activity [2,21] and recovery from stress and attention fatigue [22], respectively. As of social health, open space facilitates of social contact [23], reduces socioeconomic health inequalities [24], while ecologically, is has cooling effects [25], can reduce noise reduction [26] and filter air [27].

Taiwan is a country with a very high population density (649 persons/km^2^) and high urbanization. In such a crowded country, open space is very precious, especially in metropolis areas. Besides, the disease burden of CKD in Taiwan is an important public health concern [14,28]. However, to our best knowledge, the relationship between urban open space and human kidney function has not yet been studied. Hence, this study was conducted to explore the association between proximity to urban open space and renal function of residents.

We used logistic regression method to evaluate variables’ odds ratio for CKD. Multiple analysis results revealed that 100 m distance to open space had an odds ratio of 1.072 for CKD (Table 3). Results obtained from multiple linear regression on eGFR showed that distance to open space 100 m is associated with lower eGFR (Table 5). These findings are biologically plausible because urban open space facilitates the physical activity of residents [2,29], which in turn is beneficial to renal function [30,31]. Both proximity to green spaces [2] and built environments [29] are revealed to be associated with increased physical activity in the two review studies. Furthermore, previous studies found that better renal function is associated with physical activity. Joseph et al. [30] reviewed the data of the Third National Health and Nutrition Examination Survey (NHANES III) and found a clear association between physical activity and GFR, particularly in adults without metabolic syndrome. Another US study using the NHANES database and estimates subjects’ objective physical activity by both accelerometer and questionnaire [31]. They found a positive association of total and light physical activities with renal function. The mechanisms behind the association of physical activity with better renal function include physical activity that leads to better glycemic control in type 2 diabetes mellitus [32] and better BP control. DM and hypertension are the leading causes of CKD. Hyperlipidemia is also a risk factor of CKD and physical activity may also result in better lipid control [33].

The second reason accounting for a better renal function is that urban open space may provide a function of air filtration [34]. McPherson et al. estimated that approximately 9.8 tons of PM_10_ had been removed by trees in the Chicago area per day [35]. On the other hand, air pollution is associated with poorer renal function. A recent longitudinal study explored the association between PM_2.5_ and renal function in older men [36] and found that 1-year PM_2.5_ exposure was associated with a decrease in eGFR of 1.87 mL/min/1.73 m^2^. The possible mechanism involved includes inflammation, oxidative stress, blood pressure, and vascular/endothelial function as a result of exposure to air pollutants. With the effect of air filtration of trees, residents living near green spaces may benefit from having better air quality. However, it needs more study to confirm this relationship.

Besides distance to open space, other demographic characteristics and comorbidities of subjects were also examined for their association with renal function. Consistent with previous studies, factors significantly associated with CKD included aged, male gender, uneducated, smoking, hypertension, overweight, impaired fasting glucose (IFG), and proteinuria (Table 2). After multiple analysis of logistic regression of CKD, factors with an odds ratio for CKD included distance to open space, PM_coarse_, aged, male gender, uneducated, overweight, proteinuria, hypercholesteremia, anemia and leukocytosis (Table 3). 

To further explore the effect of distance on CKD, we performed subgroups analysis to compare certain groups of residents who live within 200 m to open space and who live more than 200 m to open space. There are several reasons that we chose 200 m as distance thresholds to access open space. Natural England’s Accessible Natural Greenspace Guidance recommends that in order to allow everyone access to natural green space, they should live no more than 300 meters (5 minutes’ walk) from home [37]. A Danish survey found that 2000 adults Danes age 18-80 had good access to and use frequently of green space if they lived less than 100 m from it [38]. Pedestrian walking speed is another concern. The walking speed is faster among younger healthy male persons. The 15_th_ percentile walking speed for older pedestrians is 0.67 m per second [39]. Taken together, most aged residents can be access to nearby open space within five minutes if they live no more than 200 m from it.

Subgroups analysis of the risk of CKD found that subjects who were female, without hypertension, and without impaired fasting glucose (IFG) have a significantly higher odds ratio of CKD if they live more than 200 m from open space than those who live less than 200 m (Table 4). In other words, the negative effects of distance to open space on renal function are more prominent among female or relatively healthier residents. One possible reason is that for subjects with comorbidities that were risk factors for CKD, the effects of distance to open space on renal function are relatively much smaller than those who without the comorbidities. Other risk factors of CKD are greater drivers of kidney function loss than the distance to open space. Another possible reason is that the medication for treating the comorbidities may interfere with the association between distance to open space and renal function. Further research is necessary to elucidate the current findings and hypotheses. Anyway, our finding reassures the important role of disease prevention of urban open spaces.

In the current study, gender differences in the relationship between open space and health were also noted in other studies. A study from Dutch explored the relationship between the self-reported health of over 10,000 people and the amount of green space in their living environment. The study found the relationship was stronger for housewives and the elderly [40]. Stafford et al. explored whether associations between neighborhood characteristics and self-rated health are different for men and women in the UK. They found that the influence of the residential environment on women’s health was greater than that on men [41]. However, Richardson et al. performed a UK-wide study to explore the gender difference of the association between urban green space and health outcome. They found that cardiovascular disease mortality was associated with urban green space in men, but not in women. The postulated reason is that women have a greater concern of safety issue about green space. Besides, their leisure time exercise was more severely attenuated by having young children than men [42]. In the current study, the gender difference in the relationship between open space and health was noted. The personal safety issue is small in New Taipei City, where the open spaces are relatively small and the population density is high. A possible explanation for the greater influence of residential environments on women’s renal function is that housewives spend more time in their neighborhood environment than their husbands do. Taiwan’s female employment rate (between 15–65 years old) was only 49.62% in 2009. On the other hand, men may have higher mobility and thus spend more time far away from home. Another possible reason is that gender difference in vulnerability to negative health impacts on renal function. 

Information of the general population of New Taipei City >30 years of age in 2009 obtained through the National Health Interview Survey and accessible at the government’s web site [43] was analyzed. Compared with the general population of New Taipei City, our subjects were more likely to be overweight (50% vs. 47%) and hypertensive (33% vs. 22%), but less likely to be smokers (11% vs. 20%), consume alcohol (37% vs. 55%), and chew betel nuts (3.5% vs. 8.3%). In terms of education level, our subjects are more likely to be uneducated (7.09% vs. 1.59%), have attended elementary or junior high school (42.71% vs. 35.24%), but less likely to have received college or graduate school education (20.34% vs. 30.49%).

This study we used education level to stand for socioeconomic status and found higher odds of CKD among uneducated subjects (Table 3). This finding is comparable to the result obtained by Mitchell et al. [24] that open space can reduce the negative effect of lower socioeconomic status on health. The incidence rate ratios (IRR) for all-cause mortality in the least and the most greenery area were 1.93 and 1.43, respectively. The effect was also noted in circulatory diseases, with IRR 2.19 in the least green area and 1.54 in the most greenery area. Similar results were also noted by Maas et al. who found a stronger relationship between urban green space and health among lower socioeconomic groups [4]. The physical activity level is lower among people with low socioeconomic status [44]. The reason behind our finding is that if these people live near open space, they have greater access to places for doing exercise. Equity is at the heart of the Sustainable Development Goal (SDG) advocated by the United Nations. Urban open space may promote healthy lives and well-being for all ages (SDG 3), reduce inequalities (SDG 10), and establish sustainable cities and communities (SDG 11).

This study explored the health effects of “open space” rather than “green space”, which is more commonly in research conducted in western countries. The reason is that there are very few green spaces in New Taipei City. Most residents are unable to easily access to green space. However, open spaces such as schoolyards are much more available. In order to enhance residents doing exercise, Taiwan enforced “National Sports Act” in 2000. The Article 7 of the Act states: “Sports facilities of all levels of educational institutions should open to the public and provide access to community citizens for sporting activities, under the pretext that it does not affect the teaching and life management of schools [45].” A national survey conducted later by the Sports Administration, Ministry of Education, Taiwan enrolled 15,361 residents revealed that the most common places of exercise are nearby the home (17%) and schools (17%) [46,47]. Many residents of the city do exercise every day in their nearby schools after the classes are over.

One of the strengths of our study is that this may be the first study to explore the relationship between open space and renal function. Second, we used a general survey to explore the health effects of local residents rather than self-rated health perception. Third, this is a large cross-sectional study in an area with high population density performed by the local government. The weakness of our study is that we had not measured subjectively or objectively the real physical activity and types of physical activity of our subjects. However, the relationship between distance to open space and physical activity had been proved by several other studies. Besides, the types of physical activity in our study subjects may be different from that of the western countries. Under the influence of traditional culture, there are many citizens of all ages play Taijiquan in their nearby open spaces such as schoolyards or parks. Some others may also play Chinese boxing, martial arts, or folk dance. The second weakness is that we had not checked the medication taken by our study subjects. Medication may modify the effect of distance to open space on renal health. Third, the serum creatinine level was measured only once. The possibility of laboratory error is a concern. Besides, we had no data about the trend of renal function over time. Further studies are necessary to explore the causal relationship and long-term health effect of urban open spaces on the disease progression of CKD.

## 5. Conclusions

In conclusion, living more distant from urban open space in New Taipei Metropolis is associated with lower eGFR and higher odds of CKD. The association is stronger in female gender or relatively healthier adults without hypertension or impaired fasting glucose (IFG). Proper urban design is thus essential in Taiwan where prevalence and incidence of CKD are very high. Further researches were necessary to explore whether proximity to open space will also be associated with the renal function of residents in suburban or rural areas.

## Figures and Tables

**Figure 1 ijerph-16-02436-f001:**
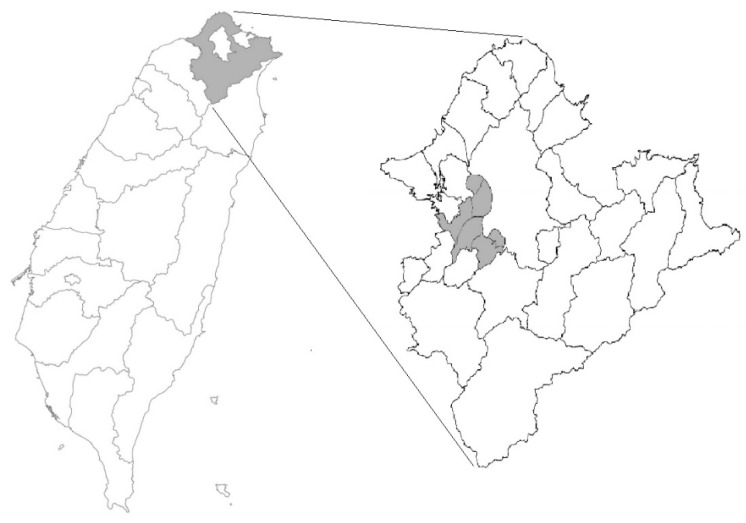
The map of Taiwan with New Taipei City in gray area (left). Residents lived in the metropolitan area of the city were included in the study (right).

**Figure 2 ijerph-16-02436-f002:**
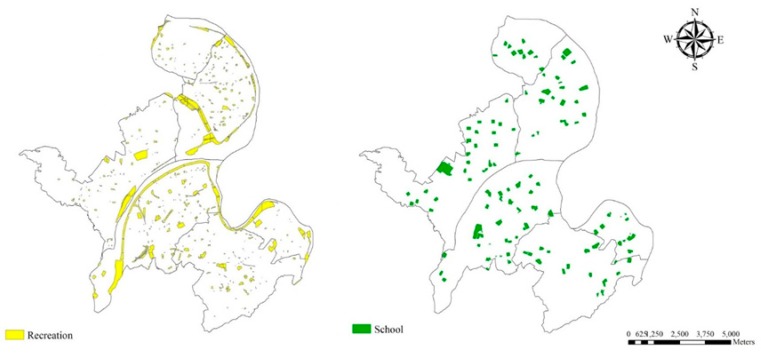
Locations of open spaces in the metropolitan area (six districts) of New Taipei City.

**Table 1 ijerph-16-02436-t001:** Characteristics of the 21,656 subjects.

Variable	Mean ± SD or N (%)
Age (years)	53.65 ± 10.37
30 ≤ Age < 40	1155 (5.33)
40 ≤ Age < 50	6670 (30.80)
50 ≤ Age < 60	8183 (37.79)
60 ≤ Age < 70	4044 (18.67)
70 ≤ Age < 80	1276 (5.89)
80 ≤ Age < 90	315 (1.45)
90 ≤ Age	13 (0.06)
Female Gender	14,477 (66.85)
Body mass index (BMI) (kg/m^2^)	24.35 ± 3.51
Waist (cm)	79.21 ± 9.97
SBP (mm Hg)	128.8 ± 20.16
DBP (mm Hg)	81.8 ± 12.15
Fasting Glucose (mg/dL)	100.61 ± 25.75
Cholesterol (mg/dL)	204.42 ± 36.54
Triglycerides (mg/dL)	118.73 ± 81.49
HDL (mg/dL)	68.96 ± 36.52
LDL (mg/dL)	113.06 ± 41.52
BUN (mg/dL)	13.48 ± 4.36
Creatinine (mg/dL)	0.84 ± 0.31
Hypertension	7164 (33.15)
Overweight	10,902 (50.44)
Impaired fasting glucose (IFG)	1550 (7.16)
Distance to Major Road (meter)	667.96 ± 453.78
Smoking	
Never	17,664 (81.62)
Former	1502 (6.94)
Current Smoker	2477 (11.44)
Alcohol consumption	
Never	13,716 (63.36)
Former	406 (1.88)
Seldom	6430 (29.70)
Current	1095 (5.06)
Ever Chew Betel nut	763 (3.53)
Education level	
Uneducated	1523 (7.09)
Elementary or junior high school	9179 (42.71)
High school	6417 (29.86)
College or graduate school	4371 (20.34)
Estimated Glomerular Filtration Rate (eGFR)	77.05 ± 13.18
Chronic Kidney Disease (CKD)	2226 (10.28)
Distance to Open Space (m)	117.23 ± 80.19
1-year exposure	
PM_2.5_ (μg/m^3^)	26.64 ± 5.01
PM_2.5Absorbance_ (10^-5^/m)	1.94 ± 0.39
PM_10_ (μg/m^3^)	49.48 ± 4.13
PM_Coarse_ (μg/m^3^)	23.13 ± 3.68

Hypertension defined as SBP ≥ 140 mmHg or DBP ≥ 90 mmHg. Overweight defined as BMI ≥ 24 (kg/m^2^). Impaired fasting glucose (IFG) defines as fasting glucose ≥ 126 mg/dL. Alcohol consumption: current defines as drinking on regular bases, seldom defines as drinking only on special occasions, formal defines as quitted from drinking on regular bases previously estimated glomerular filtration rate (eGFR) estimated by equation of chronic kidney disease (CKD)-EPI-Taiwan. CKD defined as eGFR ≤ 60 mL/min per 1.73 m^2^.

**Table 2 ijerph-16-02436-t002:** Comparison between subjects with and without CKD.

		CKD(eGFR < 60)				
		No (*n* = 19,430)		Yes (*n* = 2226)		*p*-Value
Variable		N	%	N	%	
Age	<65	17351	89.3	1004	45.1	<0.001
	≥65	2079	10.7	1222	54.9	
Gender	Female	13398	69.0	1079	48.5	<0.001
	Male	6032	31.0	1147	51.5	
Education level	Uneducated	1132	5.9	391	17.7	<0.001
	Elementary or junior high school	8048	41.7	1131	51.1	
	High school	6009	31.2	408	18.4	
	College or graduate school	4089	21.2	282	12.7	
Smoking	Never	15931	82.0	1733	77.9	<0.001
	Former	1281	6.6	221	9.9	
	Current smoker	2207	11.4	270	12.1	
Chew betel nut	Never	18720	96.5	2151	96.6	0.913
	Former	624	3.2	68	3.1	
	Current	64	0.3	7	0.3	
Hypertension	No	13280	68.5	1166	52.5	<0.001
	Yes	6111	31.5	1053	47.5	
Overweight	No	9872	50.9	838	37.7	<0.001
(BMI ≥ 24)	Yes	9518	49.1	1384	62.3	
Hypercholesteremia	No	9230	47.5	1019	45.8	0.121
(Cholesterol ≥ 200)	Yes	10198	52.5	1207	54.2	
Impaired fasting glucose (IFG) mellitus	No	18178	93.6	1928	86.6	<0.001
(AC ≥ 126)	Yes	1252	6.4	298	13.4	
Protein	-	17543	96.9	1896	88.6	<0.001
	+/-	455	2.5	140	6.5	
	+	60	0.3	50	2.3	
	++	37	0.2	24	1.1	
	+++	12	0.1	31	1.4	

*p*-value by Chi-square test.

**Table 3 ijerph-16-02436-t003:** Odds ratio for CKD.

Predictor			CKD		Multiple Analysis (Adjusted)				
		Total	N	%	Odds ratio	95% C.I.			*p*-Value
Distance to open space (100 m)	Mean ± SD	1.17 ± 0.80	1.20 ± 0.83		1.071	1.007	-	1.138	0.029
PM_Coarse_ (μg/m^3^)	Mean ± SD	23.13 ± 3.68	23.29 ± 3.76		1.017	1.003	-	1.031	0.015
Age	<65	18355	1004	5.5	1.000				
	≥65	3301	1222	37.0	6.812	6.100	-	7.607	<0.001
Gender	Female	14477	1079	7.5	1.000				
	Male	7179	1147	16.0	2.741	2.436	-	3.084	<0.001
Education	Uneducated	1523	391	25.7	1.714	1.395	-	2.104	<0.001
	Elementary or junior high school	9179	1131	12.3	1.339	1.147	-	1.562	<0.001
	High school	6417	408	6.4	1.002	0.844	-	1.189	0.983
	College or graduate school	4371	282	6.5	1.000				
Overweight (BMI ≥ 24)	No	10710	838	7.8	1.000				
	Yes	10902	1384	12.7	1.308	1.176	-	1.454	<0.001
Hypertension	No	14446	1166	8.1	1.000				
	Yes	7164	1053	14.7	1.184	1.066	-	1.315	0.002
Protein	-	19439	1896	9.8	1.000				
	+/-	595	140	23.5	2.156	1.717	-	2.707	<0.001
	+	110	50	45.5	5.645	3.593	-	8.866	<0.001
	++	61	24	39.3	3.701	1.954	-	7.010	<0.001
	+++	43	31	72.1	15.028	7.091	-	31.845	<0.001
Cholesterol (mg/dL)	Mean ± SD	204.4 ± 36.5	206.6 ± 38.8		1.002	1.001	-	1.004	0.001
RBC	Mean ± SD	4.60 ± 0.50	4.53 ± 0.57		0.533	0.474	-	0.599	<0.001
WBC	Mean ± SD	6.27 ± 1.60	6.62 ± 1.73		1.097	1.064	-	1.132	<0.001

**Table 4 ijerph-16-02436-t004:** Subgroup analyses of the effect of distance to open space on CKD.

		Distance to Open Space (m)												
		≤200			>200									
			CKD (eGFR < 60)			CKD (eGFR < 60)		Odds in >200 m Group Compared to odds in ≤200 m group						Interaction
Subgroup		Total	N	%	Total	N	%	Forest Plot	Odds ratio	95% C.I.			*p*-Value ^a^	*p*-Value ^b^
	Overall	18531	1871	10.1	3125	355	11.4	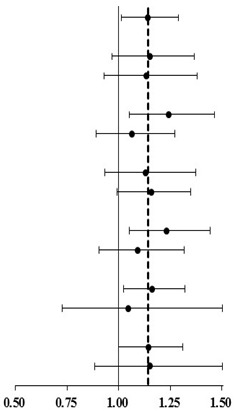	1.141	1.012	-	1.287	0.032	
Age	<65	15721	843	5.4	2634	161	6.1		1.149	0.966	-	1.367	0.117	<0.001
	≥65	2810	1028	36.6	491	194	39.5		1.132	0.930	-	1.378	0.215	
Gender	Female	12365	893	7.2	2112	186	8.8		1.241	1.052	-	1.463	0.011	<0.001
	Male	6166	978	15.9	1013	169	16.7		1.062	0.888	-	1.270	0.508	
Overweight(BMI ≥ 24)	No	9165	705	7.7	1545	133	8.6		1.130	0.931	-	1.372	0.215	<0.001
	Yes	9330	1162	12.5	1572	222	14.1		1.156	0.990	-	1.349	0.066	
Hypertension	No	12289	962	7.8	2157	204	9.5		1.230	1.050	-	1.441	0.011	<0.001
	Yes	6201	902	14.5	963	151	15.7		1.092	0.906	-	1.318	0.355	
Impaired fasting glucose (IFG) mellitus(AC≥126)	No	17193	1615	9.4	2913	313	10.7		1.161	1.022	-	1.320	0.022	<0.001
	Yes	1338	256	19.1	212	42	19.8		1.044	0.725	-	1.503	0.816	
Smoking	No	15087	1453	9.6	2577	280	10.9		1.144	0.999	-	1.310	0.052	0.008
	Yes	3432	416	12.1	547	75	13.7		1.152	0.884	-	1.501	0.294	

The dashed vertical line indicates the overall odds ratio (1.141), and the solid vertical line indicates no risk (odds ratio = 1.00). *p*-value ^a^ is from test statistics for odds of CKD between distance to open space ≤200 m group and >200 m group; *p* value ^b^ is from test statistics for interaction between distance to open space and subgroup variable.

**Table 5 ijerph-16-02436-t005:** Multiple linear regression of predictors on eGFR.

Predictor	β	SE	Std β	95% C.I. for β			*p*-Value
(Constant)	128.201	1.032		126.178	to	130.224	<0.001
Distance to open space (100 m)	−0.185	0.083	−0.012	−0.348	to	−0.021	0.027
PM_Coarse_ (μg/m^3^)	−0.122	0.018	−0.035	−0.157	to	−0.086	<0.001
Age	−0.599	0.008	−0.452	−0.614	to	−0.584	<0.001
Gender (Male vs. Female)	−4.364	0.164	−0.161	−4.685	to	−4.044	<0.001
BMI (kg/m^2^)	−0.082	0.021	−0.022	−0.123	to	−0.041	<0.001
DBP (mm Hg)	−0.025	0.006	−0.024	−0.037	to	−0.013	<0.001
Cholesterol (mg/dL)	−0.007	0.002	−0.019	−0.011	to	−0.003	<0.001
RBC	0.779	0.157	0.030	0.472	to	1.086	<0.001
WBC	−0.257	0.044	−0.032	−0.344	to	−0.170	<0.001
BUN (mg/dL)	−0.807	0.017	−0.269	−0.840	to	−0.773	<0.001
Protein	−4.770	0.330	−0.077	−5.418	to	−4.123	<0.001

β: regression coefficient; Std β: standardized regression coefficient.

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
