# Peer review of "Urban Open Space Is Associated with Better Renal Function of Adult Residents in New Taipei City"

_ijerph, 2019, doi:10.3390/ijerph16132436_

Round 1

Reviewer 1 Report

Chien et al., have performed a cross-sectional study on 21,656 patients in Taipei metropolis to explore the association between proximity to open space and adult renal function.

The study question is relevant, and the authors have done an excellent job in the design and execution of the study. The manuscript is well prepared and presented. The authors discussed the main limitations of the study, including a single measure of Cr/eGFR, lack of data on medications along with lack of data on types of physical activity.

The following are my comments,

1. Page 2 of 14.

Under Materials and methods:

Please elaborate on how alcohol consumption was rated or considered significant?

2.Page 3 of 14.

Under Statistical analysis: The authors wrote, ‘Patients were divided into two groups according to the level of eGRF: with and without CKD.’ Correct it to eGFR.

3.Page 5 of 14.

Last paragraph.

Authors mentioned ‘proteinuria’ to suggest the variables significantly associated with CKD.

Proteinuria is not mentioned earlier in the manuscript under materials and methods. Please indicate how and when it was measured and the definitions.

Author Response

The following are my point-by-point response to the reviewer’s comments:

1. Page 2 of 14.

Under Materials and methods:

Please elaborate on how alcohol consumption was rated or considered significant?

Reply: Thank you for your comments. I have added definition of drinking in footnote of Table 1. All the 4 status of drinking were used in the statistical analysis

Alcohol consumption: current defines as drinking on regular bases, seldom defines as drinking only on special occasions, formal defines as quitted from drinking on regular bases previously’

2. Page 3 of 14.

Under Statistical analysis: The authors wrote, ‘Patients were divided into two groups according to the level of eGRF: with and without CKD.’ Correct it to eGFR.

Reply: Thank you for your comment. I have changed the term in the text.

3. Page 5 of 14.

Last paragraph.

Authors mentioned ‘proteinuria’ to suggest the variables significantly associated with CKD.

Proteinuria is not mentioned earlier in the manuscript under materials and methods. Please indicate how and when it was measured and the definitions.

Reply: Thank you for your comment. I have inserted the relevant information in the materials and methods section (page 3).

‘Urinanalysis. Each subject’s first morning mid-void urine was collected during the day of survey. Urinanalysis was performed using Multistix test strips (Bayer Diarnostics, Victoria, Australia) which grade proteinuria as negative, trace, 1+ (0.3 g/L), 2+ (1 g/L), 3+ (3 g/L) or 4+ (³20 g/L) by a reaction with tetrabromphenol blue.‘

Reviewer 2 Report

In this manuscript, authors demonstrated that proximity of open space such as green space, playground, and public plaza was significantly associated with lower prevalence of chronic kidney disease (CKD) among general adult people in Taiwan. The subject of study seems to be interesting and novel. The reviewer has some comments as follows.

Major comments

The factors that mediated between proximity to open space and lower renal function remains unclear in this study. Although authors suggested two potential factors, physical activity and air pollution, causal relationship between these factors and the proximity to open space was not clearly described in the manuscript. The reviewer cannot easily agree with that people who live close to open space maintain high physical activity and avoid air pollution. Authors should show the convincing associations between these factors and the proximity to open space.

Minor comments

1. Authors should describe the approval number assigned by ethical committee.

2. The terms “diabetes” and “hyperlipidemia” seem to be inappropriate. These terms should be corrected to “impaired fasting glucose (IFG)” and “hypercholesteremia”.

Author Response

The following are my point-by-point response to the reviewer’s comments:

Major comments

The factors that mediated between proximity to open space and lower renal function remains unclear in this study. Although authors suggested two potential factors, physical activity and air pollution, causal relationship between these factors and the proximity to open space was not clearly described in the manuscript. The reviewer cannot easily agree with that people who live close to open space maintain high physical activity and avoid air pollution. Authors should show the convincing associations between these factors and the proximity to open space.

Reply: Thank you for your comment. In the current study, we did not measure the physical activity of the study subjects, so we can’t make sure that the beneficial effect is due to exercise. That’s a point of weakness that we have discussed in the last part. However, the association between proximity to open space and physical activity have been confirmed by many previous studies. So we add reference [29] and the following sentence in the article, page 10:

Both proximity to green spaces [2] and built environments [29] are revealed to be associated with increased physical activity in the two review studies.

In regard to air quality, whether the air filtration effect of trees result in better air quality of nearby residents is not clear. So we added more descriptions in page 11 and page 12:

p.11 ‘With the effect of air filtration of trees, residents living near green spaces may benefit from having better air quality. However, it needs more study to confirm this relationship.

p.12 ‘Further studies are necessary to explore the causal relationship and long-term health effect of urban open spaces on the disease progression of CKD.

Minor comments

1. Authors should describe the approval number assigned by ethical committee.

Reply: Thank you for your comment. I have provided the approval number in Line 61, page 2.

2. The terms “diabetes” and “hyperlipidemia” seem to be inappropriate. These terms should be corrected to “impaired fasting glucose (IFG)” and “hypercholesteremia”.

Reply: Thank you for your comment. I have changed all the terms in the text.

Round 2

Reviewer 2 Report

Authors have addressed the reviewer's concerns in the revised manuscript.